# Efficacy and Safety of the Combination of Superoxide Dismutase, Alpha Lipoic Acid, Vitamin B12, and Carnitine for 12 Months in Patients with Diabetic Neuropathy

**DOI:** 10.3390/nu12113254

**Published:** 2020-10-23

**Authors:** Triantafyllos Didangelos, Eleni Karlafti, Evangelia Kotzakioulafi, Zisis Kontoninas, Charalampos Margaritidis, Parthena Giannoulaki, Konstantinos Kantartzis

**Affiliations:** 1Diabetes Center, 1st Propaedeutic Department of Internal Medicine, Medical School, University General Hospital of Thessaloniki AHEPA, Aristotle University of Thessaloniki, 54636 Thessaloniki, Greece; linakarlafti@hotmail.com (E.K.); evelinakotzak@hotmail.com (E.K.); drziko@gmail.com (Z.K.); babismarg14@yahoo.gr (C.M.); 2Department of Nutrition and Dietetics, University General Hospital of Thessaloniki AHEPA, 54636 Thessaloniki, Greece; nenagian@yahoo.com; 3Department of Internal Medicine IV, Division of Endocrinology, Diabetology and Nephrology, University of Tübingen, 72076 Tübingen, Germany; Konstantinos.Kantartzis@med.uni-tuebingen.de; 4Institute for Diabetes Research and Metabolic Diseases (IDM) of the Helmholtz Centre Munich at the University of Tübingen, 72076 Tübingen, Germany; 5German Center for Diabetes Research (DZD), 72076 Tübingen, Germany

**Keywords:** measurement, peripheral neuropathy, diabetes mellitus, neurometrix

## Abstract

Aim: To investigate the efficacy of Superoxide Dismutase, Alpha Lipoic Acid, Acetyl L-Carnitine, and Vitamin B12 (B12) in one tablet in Diabetic Neuropathy (DN). Patients–methods: In this prospective, double-blind, placebo-controlled study, 85 patients with Diabetes Mellitus Type 2 (DMT2) were randomly assigned, either to receive the combination of four elements (active group, *n* = 43), or placebo (*n* = 42) for 12 months. We used the Michigan Neuropathy Screening Instrument Questionnaire and Examination (MNSIQ and MNSIE), measured the vibration perception threshold (BIO), and Cardiovascular Autonomic Reflex Tests (CARTs). Nerve function was assessed by DPN Check [sural nerve conduction velocity (SNCV) and amplitude (SNAP)]. Pain (PS) and quality of life (QL) questionnaires were administered. Results: At follow-up, BIO, MNSIQ, QL, PAIN, and SNCV, SNAP, and B12 levels had significantly improved inactive group (*p* < 0.001, *p* < 0.001, *p* < 0.001, *p* < 0.001, *p* = 0.027, *p* = 0.031, and *p* < 0.001 respectively), whereas the inplacebo group MCR (mean circular resultant) and PAIN deteriorated (*p* < 0.001, *p* < 0.001). The changes in MNSIQ, QL, SNCV, BIO, and PAIN differed significantly between groups (*p* < 0.001, *p* < 0.001, *p* = 0.031, *p* < 0.001, and *p* < 0.001 respectively). Conclusions: The combination of the four elements in one tablet for 12 months in patients with DMT2 improved all indices of peripheral neuropathy, including SNAP and SNCV, pain, and Quality of Life perception, except CARTs and MNSIE.

## 1. Introduction

One of the most common and costly microvascular complications of diabetes mellitus (DM) is diabetic neuropathy (DN), which, of note, is usually underdiagnosed and undertreated in every day clinical practice. The most common forms of DN are diabetic distal symmetrical sensorimotor polyneuropathy (DPN) and diabetic autonomic neuropathy (DAN) [1,2]. At least 50% of patients with Diabetes Mellitus Type 2 (DMT2) and up to 59% of DM type 1 develop DPN, whereas Diabetic Cardiovascular Autonomic neuropathy (DCAN) has a prevalence ranging from 2.5 to 50% [3]. DCAN and DPN are major risk factors for foot ulcers, amputation, and cardiovascular dysfunction [4]. Approximately 30% of patients develop the so-called Painful Diabetic Neuropathy (PDN) [5] with neuropathic pain and symptoms like burning, needles, cold, or heat cramps, electric shock, weakness, and allodynia [6,7,8]. Due to the painful symptoms and resulting disabilities, DN strongly affects quality of life [4,9,10]. The management of DN, and especially of PDN is a challenge for the medical community [4,10,11,12,13].

Strict glycemic control is shown to delay the progression of DN in DMT1 patients [10], but in DMT2 patients the effects are modest [14]. Data from large studies show little or no effect in DPN [15,16,17], but a reduction in the progression of CAN [15]. Others showed improvement in microvascular lesions in nerves and neuropathy parameters [18,19]. However, even these studies do not agree on the optimum level of glycemic control for preventing neurophysiological deterioration in DMT2. Furthermore, strict glycemic control must be maintained for 3–5 years for clinical benefit [20].

The distinct mechanisms underlying the development of DN in DM patients remain unknown [21], but long-standing hyperglycemia is the main key factor because of the metabolic alterations it induces, such as increased polyol flux, accumulation of advanced glycation end products, oxidative stress, and lipid disorders [22]. The duration of DM and cardiovascular risk factors (hyperlipidemia, smoking, hypertension, and obesity) also play a significant role. Alterations of microvessels, similar to those observed in diabetic retinopathy and nephropathy appear to be associated with the pathologic alterations of nerves [22,23].

No single drug proved to be effective enough for DN treatment [24]. Several drugs have been suggested but without high efficacy, such as aldose reductase inhibitors, angiotensin converting enzyme (ACE) inhibitors, c-peptide, a-lipoic acid, and benfothiamine or drugs targeting pain, such as anticonvulsants, pregabalin, duloxetine, tapedantol, antidepressants, and opioids [4,25,26,27]. This suggests that targeting more than one of the above mentionedmechanisms involved in the pathogenesis of DN may be a more effective strategy.

Superoxide Dismutase (SOD) and A-Lipoic acid (ALA) exert an antioxidant action, SOD by preventing formation of free radicals [28], and ALA by removing already formed free radicals [29]. N-acetyl-carnitine (ALC) is believed to have a neurotrophic action [30]. In addition, vitamin B12 (B12) levels are often low in DM patients, due to metformin treatment as the first-line agent of diabetes treatment [31] and is also commonly for the use of proton pumps inhibitors (PPIs) [32]. In particular, metformin is associated with a low concentration of B12 and a higher risk of B12 deficiency, but the exact mechanism remains elusive [33,34].

We therefore undertook the present study to investigate the efficacy of a new combination of these four elements, SOD 10mg daily, ALA 570 mg daily, ALC 300mg daily, and B12 250mcg daily, contained in one tablet, in DN in patients with DMT2, and generalized neuropathy (both DPN and DCAN) who have received metformin for at least four years. We hypothesized that by acting at different metabolic key points these four elements could work synergistically and thereby be more effective in treating DN in patients with DMT2.

## 2. Materials and Methods

### 2.1. Patients Recruitment

Over an 18-month period(September 2017 to February 2019), 85 consecutive adult patients with DM type 2 who fulfilled the following criteria were enrolled in the study: (a) Regular attendance (every 3 months) of the outpatient diabetes clinic of our hospital to ensure the systematic monitoring of glycemic control and diabetic complications, (b) diabetes duration and metformin treatment for at least 4 years, (c) established peripheral and autonomic neuropathy diagnosed according the criteria of Toronto panel for autonomic neuropathy (two or more Cardiovascular Autonomic Reflex Tests (CARTs)abnormal) [35], Michigan Neuropathy Screening Instrument Questionnaire (MNSIQ)/Examination (MNSIE)scoring algorithm for peripheral neuropathy (result ≥2 for lower extremity examination, ≥7 for questionnaire, and ≥2.5 for MNSI examination were considered abnormal) [36], and abnormal nerve conduction velocity with abnormal MNSIQ and MNSIE [2], and (d) acceptable good glycemic control (glycated hemoglobin (HbA1c) 6.5–7.5% or 48–58 mmol/L) [37]. Patients with clinical evidence of severe or acute cardiovascular disease (myocardial infraction or stroke in the last year), atrial fibrillation, or other cardiac arrhythmias, renal disease, cerebrovascular disease, and any other systematic disease were excluded.

### 2.2. Randomization and Allocation

The patients (41 women and 44 men) were randomized in 2 groups: Group A, *n* = 43, received the tablet with the combination of the 4elements (SOD, ALA, B12, and ALC) (Combinerv; Libytec Pharmaceutical Company SA, Athens, Greece) and group B, *n* = 42, who were given placebo for 12 months (see Appendix A).

For the allocation of participants, a random sequence of numbers by a computer program for randomization of patients into two groups of treatment was generated. The researcher who provided the randomization order and the statistician was unaware of the participants and type of treatment given in each group. The tablet with the combination did not differ from placebo and was in a plain package without any marks. The physician who provided the supplements to patients and who performed all tests and measurements was blind to the type of treatment in each group and the group each patient was allocated to. All patients completed the study (no drop-out).

### 2.3. Antidiabetic and Concomitant Medication

All patients were on treatment either with a combination of metformin and other antidiabetic drugs (Dipeptidyl peptidase 4 (DPP4) inhibitors, Glycagon-like peptide-1 (GLP-1) agonists, and Sodium glucose transporters 2 (SGLT-2 inhibitors) or with a combination of metformin with insulin and other drugs (DPP4 inhibitors, GLP-1 agonists, and SGLT-2 inhibitors). No one patient received sulfonylureas. All kinds of treatment included metformin. A daily dose of metformin for all patients in both groups was ≥1500 mg. A total of 72.3% of our patients in both groups presenting hyperlipidemia (total cholesterol >200, low density lipoprotein (LDL) cholesterol >130, and high density lipoprotein (HDL) cholesterol <40) [38], received statins. Any treatment for diabetes or other disorder did not change during the 12 months of follow up. A similar percentage of patients used drugs for hypertension and cardiovascular disease in both groups of patients. With the exception of the tablet, patients did not receive any other medications for managing pain until the end of follow-up.

### 2.4. Measurements and Tests

The following methods were used for detecting DPN and Diabetic Autonomic Neuropathy (DAN): The Michigan Neuropathy Screening Instrument Questionnaire and Examination (MNSIQ and MNSIE) [36,39], measurement of vibration perception threshold with biothesiometer (BIO) (Newbury, OH, USA) [40], and an assessment of large-fiber function using the ‘DPN-Check’ (Neurometrix Inc., Waltham, MA, USA) [6,41]. Recently, a novel point-of-care sural nerve conduction device has been developed and sural nerve functions were measured using DPN Check [sural nerve conduction velocity (SNCV) and amplitude (SNAP)]. The methodology, practical application, and validation of these tests are described in details elsewhere [36,39,40,41]. All tests were performed on the same day by an experienced physician blinded to the treatment.

Cardiovascular Reflex Tests (CRT), which are considered to be the gold standard measurement, were used for screening Cardiovascular Autonomic Neuropathy [35]. CRTs were performed using HOKANSON ANS Reader. Briefly, the following CARTs were performed: Cardiovascular Reflex Tests (CRT): R-R variation during deep breathing [assessed by mean circular resultant (MCR)], Valsalva maneuver (Vals), postural index (PI) calculated as a 30:15 ratio, and blood pressure response to standing (Orthostatic hypotension (OH) or PI). Full details are described elsewhere [4]. Age-specific reference values were applied. The first two CARTs address parasympathetic function, and Valsalva evaluates both parasympathetic and sympathetic function, whereas OH assesses sympathetic integrity [42].

We also used a pain (PS) and a quality of life (QOL) questionnaire. Pain DETECT questionnaire was self-administered and addressed the quality of neuropathic pain symptoms [43] whereas for evaluating the quality of life, a 15-item DQOL Brief Clinical Inventory was used. This is a brief-focused version of the Diabetes Quality of Life (DQOL) questionnaire, which was used in Diabetes Control and Complications Trial (DCCT) [10,44].

The entire study was approved by the local ethical committee of Medical School of Aristotle University of Thessaloniki (No: 42292). All participants signed a consent form. The reported investigations have been carried out in accordance with the principles of the declaration of Helsinki. All procedures were performed early in the morning and in line with published guidelines regarding patient preparation, hydration, physical activity, and hypoglycemia [35].

### 2.5. Method of Measurement for Vitamin B12

A measurement of vitamin Β12 was performed with the method of electrochemiluminescence (ECLIA) with analyzer Cobas e 602, as described by the manufacturer [45,46,47]. The results are expressed in pg/mL. All laboratory measurements of biochemical markers took place in the central laboratory with the same analyzer.

### 2.6. Statistical Analysis

Statistical analysis was performed with the International Business Machines Corporation (IBM) Statistical Package for the Social Sciences (SPSS) program [48]. Data for continuous variables are presented as mean ± standard deviation. Continuous variables were compared between the active and placebo groups using independent samples student’s t-test and comparisons between baseline and end of intervention were performed with paired samples t-test for each group. To investigate the difference in the mean change of relevant parameters between groups, multiple general linear regression (ANCOVA) was performed adjusting for antidiabetic medication. *p* < 0.05 was considered statistically significant.

## 3. Results

At the baseline, there was no difference between the active and placebo group in regard to all demographic characteristics (Table 1), laboratory measurements, and all test results including neurophysiological parameters, as shown in Table 2 and Table 3 and in Appendix B, Table A1. The mean age at baseline was 63±11 years and the mean duration of DM was 15 years. A total of 24 patients in the active group and 22 in the placebo group received only oral treatment with metformin plus other oral antidiabetic drugs except for sulfonylureas. The rest of the patients received treatment with metformin plus basal insulin plus other oral antidiabetic drugs. The duration of receiving metformin was at least four years (active group, range 4–40 years; placebo group, range 4–33 years).

Laboratory measurements and clinical tests values (MNSIQ, MNSIE, SNAP, SNCV, BIO, MCR, PI, PO, and Valsalva) at the baseline are shown in Table 2 and Table 3. SNCV was not obtainable in 26.3% and SNAP in 22.8% of our population, indicating severe DPN. More biochemical markers at baseline are shown in the Appendix B see Table A1. All values, including HbA1c and Vitamin B12 levels, did not differ between the two groups at baseline.

During follow-up, BIO, MNSIQ, QL, PAIN, SNCV, SNAP, and B12 levels were significantly improved in the active group (Table 4). Indices of CARTs, MNSIE, HbA1c (Table 4), as well as blood pressure (Appendix B, Table A2) and laboratory measurements (blood count and lipid profile, (see Appendix B, Table A2), urea, creatinine, SGOT, SGPT, and TSH) remained unchanged (data not shown). Percentage of pain decreased statistically significantly by 16% in the active group whereas the placebo group reported that pain deteriorated significantly.

We did not observe a significant change in any of the laboratory measurements and indices of neuropathy in the placebo group including B12 and HbA1c levels, except MCR, which decreased during follow-up (Table 4) meaning a deterioration of the function of the parasympathetic nervous system.

Between the two groups there was a significant difference in the change (follow-up–baseline) of the following parameters (adjusted for antidiabetic medication): B12 (*p* = 0.018), MNSIQ (*p* < 0.001), QL (*p* < 0.001), SNCV (*p* = 0.031), BIO (*p* < 0.001), PO (*p* < 0.001, and PAIN (*p* < 0.001) (Table 4). Additional adjustment for the change in HbA1c and the change of BMI did not change the statistical significance (data not shown).

Not a single adverse event suspected or possibly related to the combination of four elements was reported.

## 4. Discussion

The present study focused on adult patients with long standing DMT2 (mean duration 15 years) with established both peripheral and autonomic, as well as painful neuropathy. According to the data published so far in the literature, hyperglycemia plays the most important role in the pathogenesis and course of DN. Therefore, most studies aiming at treating DN focused on improving glycemic control. However, data from large scale studies show that strict glycemic control in DMT2 has generally modest effects on DN. For instance, in the Steno 2 study, multifactorial risk intervention, including intensive diabetes treatment, angiotensin-converting enzyme inhibitors, antioxidants, statins, aspirin, and smoking cessation in a cohort of patients with microalbuminuria had a moderate effect on autonomic neuropathy at 4 and 8 years [16] and no effect on DPN after 7.8 years [15] and again at 13.3 years [16] of follow-up. Strict glycemic control led only to a lower rate of impaired vibration perception threshold after 15 years in the UKPDS [14], but no change in the risk of deteriorating DN after five years in the ADVANCE [17], or a not significant reduction of the incidence of neuropathy in ACCORD [49] and VADT studies [50]. Therefore, as shown in more recent studies there is a benefit of strict glycemic control in more sensitive endpoints likecorneal confocal microscopy [18] and improvement in DN parameters and metabolic control through non-erythropoietic peptide [51], after pancreas and kidney transplantation [52] and GLP-1 analogs [53]. On theother side, few recent studies showed that strict glycemic control (HbA1c < 6.5%) improved neurophysiological parameters and neuropathic symptoms [18,19]. Nevertheless, it should be noted that these studies included only small groups of patients with poorly-controlled DMT2.

Hyperglycemia is thought to act on DN by certain mechanisms, as increasing polyol flux, accumulating advanced glycation end products, increasing oxidative stress, and causing lipid disorders [22]. Therefore, in an attempt to find a more effective treatment of DN by targeting these mechanisms at multiple points, we used the combination of three elements, ALA, and SOD, which are thought to have antioxidant action [28,54,55,56,57,58,59], and ALC which has potentially neurotrophic properties [30,60,61,62]. B12 was added to the combination because patients with long-standing DM are likely to have low B12 levels [31]. This has been attributed to the chronic use of metformin and proton pump inhibitors (PPIs). Since cardiovascular disease is quite common in diabetic patients, they are often given antiplatelet therapy along with PPIs for gastric protection [32]. Furthermore, B12 deficiency has been repeatedly associated with metformin use [63,64,65]. Chronic metformin use results in vitamin B12 deficiency in about 30% of patients [63]. De Jager et al. provided the strongest evidence of metformin-associated low vitamin B12 levels in a 4.3 years randomized controlled trial (RCT). According to the findings of this study, we set the four-year minimum treatment duration for metformin as an inclusion criterion for the participants of our study [65]. Many mechanisms have been proposed to explain how metformin interferes with the absorption of B12. Intestinal bacteria overgrowth resulting in the binding of IF-vitamin B12 complex to bacteria instead of being absorbed was an early suggested mechanism [66]. It has also been proposed that metformin reduces the vitamin absorption by altering intestinal motility [67].

Each of the four elements that we used in the present study has been used as monotherapy in clinical studies to treat DN. Vitamin B12 deficiency can cause peripheral neuropathy and hematological abnormalities, such as pancytopenia [31]. In the absence of anemia, peripheral neuropathy due to B12 deficiency is often misdiagnosed as diabetic neuropathy, although clinical findings like pain or neuropathic symptoms varied. Failure to diagnose the cause of neuropathy will result in the progression of central and/or peripheral neuronal damage. A subsequent vitamin B12 replacement may arrest the further progress of nerve damage, but will not lead to its reversal [63]. Vitamin B12 supplementation may be beneficial in DPN, since it has shown to have a relieving effect on pain and improve nerve conduction and quality of life [68,69]. Mizukami et al. suggested that a correction of impaired neural signal of protein kinase C and oxidative stress-induced damage may play a major role in the beneficial effects of methylcobalamin on neuropathy in diabetic animals [70].

Acetyl L-Carnitine is the acetyl ester of L-carnitine which is synthesized in the human brain, liver, and kidney from lysine and methionine, or is taken-up through the diet [30,71]. In addition, it has been proposed for the treatment of DN due to its analgesic action mediated by reducing the concentration of glutamate in synapses [30]. It has demonstrated cytoprotective antioxidant and antiapoptotic effects in the nervous system [60,72]. In a meta-analysis of four RCTs with 523 patients, ALC was shown to significantly reduce neuropathic pain [62]. ALC 500 mg or 1000mg three times per day improved pain estimation and vibration perception threshold after one year, both in DMT1 and DMT2 patients. The dose of 500 mg was shown to increase fiber numbers and regenerate nerves to a significantly higher extent compared to 1000mg, but without improving velocity and amplitude [30]. In another study, ALC had beneficial action to DPN over 24 weeks, with an equal strong effect as methylocobalamin [73].

SOD is a metalloenzyme, present in three isoforms: Cytosolic (SOD1), mitochondrial (SOD2), and extracellular (SOD3) [28] and appears to have a potential antioxidant action through the neutralization of superoxide. Superoxide is thought to be overproduced due to hyperglycemia and leads to oxidative stress, which in turn contributes to the development of DSPN [28]. Experimental data show an association between low levels of SOD and the progression of DN [58]. SOD3 levels that are produced as a response to inflammatory cytokines [59] are inversely associated with premature sensory and motor nerve conduction changes in patients with DM type 1 and recently diagnosed DM type 2 with DSPN [58].

When orally administrated, SOD is deactivated in the gastrointestinal tract. Therefore, in the tablet provided in our study, we used the combination of SOD with the wheat gliadin biopolymer system (GliSODin^®^) to protect SOD through its passage from gastrointestinal system and improve its delayed release. The wheat gliadin increases the permeability of the intestine by promoting the increase of zonulin receptors [74], thereby allowing the transport of the macromolecule SOD across the intestinal barrier [75].

ALA is a medium chain fatty acid, naturally derived from plants and animals, Ref. [76] with potential antioxidant action and effective in the management of microvascular complications in diabetic rats, such as neuropathy, retinopathy, and other vascular disease [33]. The Neurological Assessment of Thioctic Acid in Diabetic Neuropathy trial (NATHAN) 1, the longest and largest multi-center, randomized, double-blind, parallel-group trial in patients with uncontrolled diabetes and DPN, showed a favorable effect of a daily dose of 600 mg ALA orally for four years on neuropathic symptoms but not in conduction velocity [77,78]. In the a-Lipoic Acid in Diabetic Neuropathy trial (ALADIN II) improvement in motor and sensory nerve conduction velocity and amelioration of neuropathic symptoms were found after two years of initially intravenous and then oral administration of ALA [79]. In a meta-analysis, by Mijnhout et al. [80] of ALADIN, SYDNEY (Symptomatic Diabetic Neuropathy Trial) [81], SYDNEY 2 (Symptomatic Diabetic Neuropathy Trial 2) [82], and OPRIL (Oral Pilot), 600 mg ALA either orally or intravenously was shown to reduce significantly neuropathic symptoms [78,80,81,82,83]. ALA reduced neuropathic symptoms in as only as 40 days in one study, which however did not include a control group [84]. ALA was also shown to improve glucose homeostasis and lipid profile, (except HDL cholesterol) in a recent meta-analysis of 24 RCTs [81,82,85]. Of note, the primary outcome of most of the aforementioned studies was improvement in a total symptom score and not in neurophysiological measurements. Taken together, these data suggest that ALA may improve neuropathic symptoms and in a few trials neurophysiological parameters in DN [77,78,79]. Considering that neuropathic pain is difficult to treat, and standard analgesics are usually not effective enough, ALA seems to be a promising agent to improve neuropathic symptoms and therefore quality of life, and even to reduce the use of rescue drugs, such as pregabalin, duloxetine, and tapedantol, which are commonly used by patients with diabetic neuropathy [11,86]. However, the therapeutic efficacy of orally administrated ALA may be limited by its pharmacokinetic profile. Data show a short half-life and bioavailability of about 30% of ALA due to mechanisms involving hepatic degradation, reduced solubility, as well as instability in the stomach. Nevertheless, studies have shown that ALA bioavailability is enhanced by the use of amphiphilic matrices, which are able to increase its solubility and intestinal absorption [11,87,88].

We therefore used in our study the combination of all these elements in a single tablet with the specific pharmacological preparation technology, Multiform Administration Timed Release Ingredients System (M.A.T.R.I.S.^®^) retard to ensure the best bioavailability. This technology allows the uniform and rapid distribution of the active ingredients throughout the gastrointestinal tract and the smooth release of the controlled release. Indeed, the M.A.T.R.I.S.^®^ retard technology solves the difficulties that associate with ALA and SOD absorption and bioavailability. Moreover, it gives the possibility to overlap the bitter taste of ALA [89].

Apart from the investigational tablet, we also set strict inclusion criteria and carefully chose the participants. Considering the key role of hyperglycemia in the pathogenesis and course of DN, we only included diabetic patients with stable and fair acceptable glycemic control (HbA1c 6.5–7.5%, 48–58 mmol/L). We further included only patients under metformin treatment for at least four years continuously prior to participation to the study because, as discussed above, these patients are likely to have had low B12 levels.

In this way we were able to show that administration of the combination of the four elements (ALA, SOD, B12, and ALC) in one tablet, for 12 months led to an improvement in neurophysiological parameters, as expressed by conduction velocity and amplitude of sural nerve, in vibration perception threshold, in Quality of Life, and in the MNSI Questionnaire. We further found that for the combination ameliorates neuropathic symptoms, pain was reduced approximately by 16%. Indices of CARTs remained unchanged. MNSIE improved, but not significantly. Levels of B12 increased significantly in the active group (263.0 at baseline vs 335.0 pg/mL at follow-up, *p* < 0.001), but they did not reach the level of 450 pg/mL, which has been considered the lower normal limit for avoiding neurologic dysfunction, especially in elderly people [90]. Between two two groups there was a significant difference in the change of SNCV, MNSQ, QL, BIO, and PAIN (Table 4), thus confirming the positive effect of the combination on DN. Of particular note, the most probable confounding factors, HbA1c, body weight, and treatment for diabetes remained unchanged in both groups during the 12 months of follow up, thus further strengthening the robustness of our findings. It should be noted that none of our patients presented with plantar ulcer or needed amputation but this could not be attributed to the combination used and probably glycemic control is the preventative factor as also seen in the Intensified Multifactorial Intervention in Patients with Type 2 Diabetes and Microalbuminuria (STENO) study [16].

According to our best knowledge, this is the first placebo-controlled randomized, double-blind trial of a tablet containing the combination of ALA, SOD, B12, and ALC as a food supplement that investigated the efficacy of the tablet on all three major forms of chronic generalized DN, i.e., peripheral sensorimotor neuropathy, autonomic neuropathy, and painful neuropathy.

Three studies already published in the literature investigated the effectiveness of the combination of two of our four elements in DN [91,92,93]. Up to now, only a recent study evaluated the efficacy of B12 and ALA on DPN symptoms and revealed that ALA improved burning and pain, while B12 ameliorated numbness and paresthesia [92]. A combination of B12 and ALA is superior than monotherapy of either agent in improving nerve conduction velocity and relief of neuropathic symptoms [91,93]. Nonetheless, the combination of SOD and ALA acts synergistically to reduce pain and improve nerve conduction velocity [94].

Compared to other studies [77,79,81,82], the patients in our study had relatively good glycemic control at the beginning of the trial and maintained good control during the entire follow-up period, whereas in most other studies glycemic control was poor at the baseline and improved to suboptimal level during the course of the study [77,78,79,81,82]. It follows, that in these studies, the pure efficacy of the investigated drug or substance on DN remained somehow uncertain [80,91]. Another major limitation of these studies was that they did not examine concurrently in the same patient DPN, DAN, and neuropathic pain [77,79,81,82], as we did. Of course, our study also has limitations. All patients originated from one diabetes center. This ensured homogeneity of the sample cohort, but does not allow for extrapolation of the results to other populations. We only measured sural nerve and not other nerve conduction velocity and amplitude, but it is widely accepted that sural nerve velocity is also representative of other nerves [41,95,96]. Finally, we did not measure methylomalonic acid and homocysteine, although they are associated with vitamin b12 deficiency because these are more specific and expensive markers are not always available in every-day clinical practice and are shown to not be necessarily needed for the diagnosis of vitamin B12 deficiency [97,98].

In conclusion, in the present study, the administration of the combination of the four elements in one tablet for 12 months in patients with DMT2 resulted in an improvement in all indices of peripheral neuropathy including neurophysiological parameters, pain, and quality of life, except CARTS and MNSIE, whereas in the placebo group, a deterioration of parasympathetic function was noted.

## Figures and Tables

**Table 1 nutrients-12-03254-t001:** Demographic characteristics of the study population at the baseline.

	Active (Group A)	Placebo (Group B)	*p*
Gender (m/w)	23/20	21/21	0.901
Age (years)	65.12 ± 11.16	62.34 ± 11.32	0.323
Height (cm)	170.27 ± 8.64	165.44 ± 11.7	0.062
Body weight (kg)	89.85 ± 12.5	85.22 ± 15.27	0.185
Body Mass Index (BMI) (kg/m^2^)	30.99 ± 3.78	31.19 ± 4.99	0.859
Diabetes Duration (years)	15.01 ± 8.2	14.6 ± 9.7	0.824
Metformin + other oral antidiabetic drugs (%)	24 (55.8)	22 (53.4)	0.718
Duration of receiving Metformin (years)	14.5 ± 8.5	13.2 ± 9.4	0.554
Smoking (%)	14 (32.5)	14 (35.7)	0.724
Cardiovascular disease (%)	15 (34.9)	13 (30.9)	0.633
Hyperlipidemia (%)	32 (74.4)	30 (71.4)	0.710
Hypertension (%)	28 (65.1)	28 (66.7)	0.847

Data are given as means +/− standard deviation or n (%).

**Table 2 nutrients-12-03254-t002:** Laboratory measurements at the baseline for both groups.

	Active	Placebo	*p*
HbA1c (%)	7.14 ± 1.14	6.84 ± 0.83	0.242
HbA1c(mmol/L)	54.6 ± 12.5	52.12 ± 8.67	0.240
Vitamin B12 (pg/mL)	262.9 ± 106.6	293.47 ± 125.6	0.885
Hematocrit (%)	39.65 ± 5.86	40.22 ± 3.12	0.109
Mean Corpuscular Volume (μm^3^fl)	84.72 ± 4.08	84.97 ± 9.47	0.348
Urea (mg/dL)	35.54 ± 11.26	43.55 ± 22.29	0.071
Creatinine (mg/dL)	0.90 ± 0.22	0.91 ± 0.22	0.937
Cholesterol (mg/dL)	169.48 ± 44.23	171.4 ± 50.17	0.875
Triglycerides (mg/dL)	152.55 ± 69.78	152.4 ± 56.1	0.993
HDL-cholesterol (mg/dL)	45.71 ± 12.9	49.33 ± 10.7	0.236
LDL cholesterol (mg/dL)	93.3 ± 43.2	91.6 ± 50.2	0.889

Abbreviations: HbA1c (glycated hemoglobin), HDL-cholesterol (high density lipoprotein), LDL-cholesterol (low density lipoprotein).

**Table 3 nutrients-12-03254-t003:** Indices of Cardiovascular Autonomic Reflex Tests (CARTs) and neuropathy test at baseline.

	Active	Placebo	*p*
MNSIQ	6.76 ± 1.66	6.16 ± 2.35	0.244
MNSIE	4.26 ± 3.01	3.75 ± 2.5	0.444
QL	39.83 ± 10.82	38.01 ± 10.5	0.763
SNAP (IV)	5.62 ± 5.79	6.93 ± 6.19	0.414
SNCV(m/s)	30.8 ± 23.5	38 ± 21.67	0.237
BIO (V)	33.7 ± 11.6	29.8 ± 11.9	0.192
MCR	17 ± 19.6	19.3 ± 22.4	0.660
PI	3.2 ± 1.5	2.75 ± 1.1	0.177
PO (mm/Hg)	10.5 ± 8.9	10.3 ± 10.3	0.929
Valsalva	1.50 ± 0.29	1.53 ± 0.24	0.753
Pain score	19.5 ± 7.8	16.6 ± 8.7	0.168

Abbreviations: MNSIQ (Michigan Neuropathy Screening Instrument Questionnaire), MNSIE (Michigan Neuropathy Screening Instrument Examination), QL (Diabetes Quality of Life Questionnaire), SNAP (sural nerve action potential (amplitude) in right foot), SNCV (sural sensory nerve conduction velocity in right foot), BIO (Biothesiometer), MCR (Mean Circular Resultant), PI (Postural Index), Postural hypotension PO orthostatic hypotension PAIN (Pain Score Questionnaire).

**Table 4 nutrients-12-03254-t004:** Changes in indices from baseline to end of intervention for both groups.

	Active			Placebo			*p* ^c^
	Baseline	12Months	*p* ^a^	Baseline	12Months	*p* ^b^	
HbA1c (%)	7.14 ± 1.14	6.91 ± 0.79	0.120	6.84 ± 0.83	6.88 ± 0.75	0.81	0.486
HbA1c (mmol/L)	54.6 ± 12.5	51.4 ± 9.1	0.120	52.12 ± 8.67	51.75 ± 8.19	0.806	0.486
B12 (pg/mL)	262.94 ± 106.6	335.42 ± 107.59	0.000	293.47 ± 125.6	305.1 ± 145.2	0.57	0.018
MNSIQ	6.76 ± 1.66	6.16 ± 1.7	0.000	6.16 ± 2.35	6.35 ± 2.26	0.06	<0.001
MNSIE	4.26 ± 3.01	4.06 ± 2.49	0.161	3.75 ± 2.5	3.86 ± 2.4	0.16	0.063
QL	39.83 ± 10.82	36.2 ± 9.8	0.000	38.1 ± 10.5	38.8 ± 9.7	0.21	<0.001
SNAP (IV)	5.62 ± 5.79	7.55 ± 6.58	0.031	6.93 ± 6.19	6.56 ± 6.27	0.70	0.065
SNCV (m/s)	30.8 ± 23.5	39.4 ± 22.8	0.027	38 ± 21.67	38.9 ± 21.4	0.87	0.031
BIO (V)	33.7 ± 11.6	23.3 ± 11.5	0.000	29.8 ± 11.9	28.5 ± 11.5	0.78	<0.001
MCR	17 ± 19.6	16.4 ± 13.9	0.630	19.3 ± 22.4	12.5 ± 19.1	0.01	0.220
PI	3.2 ± 1.5	3.4 ± 1.2	0.330	2.75 ± 1.1	2.9 ± 1.1	0.18	0.494
PO (mm/Hg)	10.5 ± 8.9	8,7 ± 7.3	0.001	10.3 ± 10.3	12.9 ± 9.8	0.06	<0.001
Valsalva	1.50 ± 0.29	1.55 ± 0.34	0.110	1.53 ± 0.24	1.49 ± 0.29	0.220	0.393
PAIN	19.5 ± 7.8	17.4 ± 7	0.000	16.6 ± 8.7	17.7 ± 9.4	0.00	<0.001

^a^ for difference in active group, ^b^ for difference in placebo group, ^c^ for difference between groups. Abbreviations: HbA1c (glycated hemoglobin), MNSIQ (Michigan Neuropathy Screening Instrument Questionnaire), MNSIE (Michigan Neuropathy Screening Instrument Examination), QL (Diabetes Quality of Life Questionnaire), SNAP (sural nerve action potential (amplitude) in right foot), SNCV (sural sensory nerve conduction velocity in right foot), BIO (Biothesiometer), MCR (Mean Circular Resultant), PI (Postural Index), Postural hypotension PO (orthostatic hypotension), PAIN(Pain Score Questionnaire).

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
