# Peer review of "Efficacy and Safety of the Combination of Superoxide Dismutase, Alpha Lipoic Acid, Vitamin B12, and Carnitine for 12 Months in Patients with Diabetic Neuropathy"

_nutrients, 2020, doi:10.3390/nu12113254_

Round 1

Reviewer 1 Report

This is an important contribution as it is a placebo controlled trial in DPN and DAN which is actually positive.

The intervention is of course a cocktail which limits an assessment of the major site of action for the observed benefits.

  1. The authors should acknowledge studies showing the benefit of multifactorial intervention on autonomic neuropathy (Gaede P, Lund-Andersen H, Parving HH, et al. Effect of a multifactorial intervention on mortality in type 2 diabetes. N Engl J Med 2008;358(6):580-91.) but not somatic neuropathy as only VPT was assessed. But that more recent studies have shown a benefit when using more sensitive end points such as corneal confocal microscopy for an improvement in risk factors: Ishibashi F, Taniguchi M, Kosaka A, et al. Improvement in Neuropathy Outcomes With Normalizing HbA1c in Patients With Type 2 Diabetes. Diabetes Care 2019;42(1):110-18; glucose control: Azmi S, Jeziorska M, Ferdousi M, et al. Early nerve fibre regeneration in individuals with type 1 diabetes after simultaneous pancreas and kidney transplantation. Diabetologia 2019;62(8):1478-87; nonerythropoietic peptide: Brines M, Dunne AN, van Velzen M, et al. ARA 290, a nonerythropoietic peptide engineered from erythropoietin, improves metabolic control and neuropathic symptoms in patients with type 2 diabetes. Mol Med 2015;20:658-66; GLP-1 or insulin: Ponirakis G, Abdul-Ghani MA, Jayyousi A et al. Effect of treatment with exenatide and pioglitazone or basal-bolus insulin on diabetic neuropathy: a substudy of the Qatar Study. BMJ Open Diabetes Res Care. 2020 Jun;8(1):e001420.
  2. The Toronto criteria which relies on neurological deficits alone is only for the possible or probable diagnosis of DPN as per the current study (MNSIQ and MNSIE). For confirmed DPN The presence of an abnormality of nerve conduction and a symptom or symptoms or a sign or signs of neuropathy are required.
  3. It seems like an unbelievable study with no change in medications for the duration of the study and also no single patient needed to take medication to relieve neuropathic pain.
  4. What happened to lipids, BP etc. during the study?
  5. The SNCV at baseline is very low as Neurometrix normally overestimates this and therefore these patients appear to have a very severe DPN. What proportion of patients was SNCV and SNAP not obtainable?
  6. Most surprising is that despite SNCV of 30.8 m/s at baseline they improved by a remarkable and unheard of 9 m/s! This has not even been seen after pancreas and kidney transplantation.
  7. Similarly VPT improved by 10V! again this challenges the results of the STENO with multiple risk factor reduction achieving very little change in VPT.
  8. How do you explain the normal B12 levels in your population given that <150 indicates deficiency?

Reviewer 2 Report

This is a very interesting study focusing in a novel approach to diabetic neuropathy. I have no important suggestions, but it would be interesting (as neuropathy leads to plantar ulceration) comment the possible effect of the experimental treatment in avoiding plantar ulcers, including a citation.
